# Biofilm Formation Plays a Crucial Rule in the Initial Step of Carbon Steel Corrosion in Air and Water Environments

**DOI:** 10.3390/ma13040923

**Published:** 2020-02-19

**Authors:** Akiko Ogawa, Keito Takakura, Nobumitsu Hirai, Hideyuki Kanematsu, Daisuke Kuroda, Takeshi Kougo, Katsuhiko Sano, Satoshi Terada

**Affiliations:** 1Department of Chemistry and Biochemistry, National Institute of Technology (KOSEN), Suzuka College, Suzuka 510-0294, Japan; takakura.0820@gmail.com (K.T.); hirai@chem.suzuka-ct.ac.jp (N.H.); 2Department of Material Science and Engineering, National Institute of Technology (KOSEN), Suzuka College, Suzuka 510-0294, Japan; kanemats@mse.suzuka-ct.ac.jp (H.K.); daisuke@mse.suzuka-ct.ac.jp (D.K.); kougo@mse.suzuka-ct.ac.jp (T.K.); 3D&D Company, Yokkaichi 512-1211, Japan; sano@ddcorp.co.jp; 4Department of Materials Science and Biotechnology, University of Fukui, Fukui 910-8507, Japan; terada@u-fukui.ac.jp

**Keywords:** thermal spray, corrosion, zinc, copper, steel

## Abstract

In this study, we examined the relationship between the effect of a zinc coating on protecting carbon steel against biofilm formation in both air and water environments. SS400 carbon steel coupons were covered with a zinc thermal spray coating or copper thermal spray coating. Coated coupons were exposed to either air or water conditions. Following exposure, the surface conditions of each coupon were observed using optical microscopy, and quantitatively analyzed using an x-ray fluorescence analyzer. Debris on the surface of the coupons was used for biofilm analysis including crystal violet staining for quantification, Raman spectroscopic analysis for qualification, and microbiome analysis. The results showed that the zinc thermal spray coating significantly inhibited iron corrosion as well as biofilm formation in both air and water environments. The copper thermal spray coating, however, accelerated iron corrosion in both air and water environments, but accelerated biofilm formation only in a water environment. microbially-influenced-corrosion-related bacteria were barely detected on any coupons, whereas biofilms were detected on all coupons. To summarize these results, electrochemical corrosion is dominant in an air environment and microbially influenced corrosion is strongly involved in water corrosion. Additionally, biofilm formation plays a crucial rule in carbon steel corrosion in both air and water, even though microbially-influenced-corrosion-related bacteria are barely involved in this corrosion.

## 1. Introduction

Zinc coatings are widely used in building products for the anti-corrosive treatment of steel. Zinc coatings mainly protect steel against corrosion through barrier and galvanic protection effects [1]. In general, a metallized zinc coating can produce a thicker layer than a galvanized one [2], which guarantees beneficial effects for resistance to corrosion. The thermal coating of zinc is one type of metallic zinc coating, which is commonly applied to bridges and marine structures where atmospheric corrosion is likely to be accelerated by soluble salts such as sodium chloride [3].

Biofilm formation also enhances the metallic corrosion of steels [4]. Biofilms refer to microbially produced three-dimensional structures that consist of water (over 80%), microbes, and extracellular polymeric substrates (EPSs) produced by the microbes [5]. Biofilm-related corrosion is also referred to as biocorrosion or microbially influenced corrosion (MIC). Several mechanisms for MIC have been proposed, as follows: galvanic cells, differential aeration or chemical concentration cells, acidity of organic acids derived from bacterial metabolites, secretion of H_2_S, NH_3_, or PH_3_ from bacteria, enzymatic oxygen reduction reactions, and direct extraction and consumption of electrons from iron [6].

Many studies have investigated either electrical corrosion or MIC [7,8,9,10,11,12,13,14,15], however, none have dealt with a combination of them. In this study, we investigated a corporate-steel corrosion mechanism that connects the gap between electrical corrosion and MIC under both air atmosphere and water-immersion conditions. At the same time, we examined the inhibitory effect of the zinc thermal spray coating on steel corrosion in comparison with a copper thermal spray coating. The reason why we tested the copper thermal spray coating was that we expected that copper would postpone and (or) regulate MIC. Some researchers have reported that copper can inhibit microbial activities of some bacteria [16,17,18]. Additionally, we found that copper downregulated the biofilm formation of marine living bacteria [19]. In order to elucidate the steel corrosion mechanism, we analyzed the corroded test sites by microscopic observation and energy-dispersive x-ray (EDX) analysis; we also analyzed any biofilms formed using a combination of quantitative and qualitative methods to explore the microbiome.

## 2. Materials and Methods

### 2.1. Thermal Sprayed Coupons

JIS SS400 carbon steel plate coupons (10 cm × 10 cm, thickness 1 mm, Sakai Netsu-Giken, Tsu, Japan) were subjected to a zinc or copper coating using a wire flame spraying process at TOCALO Co. Ltd. (Kobe, Japan). The thickness of the sprayed layer was about 1 mm. For the remainder of this manuscript, we refer to SS400 carbon steel plate, zinc thermal spray-coated SS400 carbon steel plate, and copper thermal spray-coated SS400 carbon steel plate, as SS400, Zn-coated, and Cu-coated, respectively.

### 2.2. Outdoor Exposure Test

Each coupon was put on an acrylic plate (Sakai Netsu-Giken) with two corners held in place by stainless steel clamps (Sakai Netsu-Giken). The acrylic plates were fixed on a wooden board at a distance of 5 cm from the board (Figure 1). This board was then left on the rooftop deck of a building (10 m tall) at the National Institute of Technology, Suzuka College (34.8502 N, 136.58132 E), located 2 km from the Shiroko coast, for two months (from 1 December 2016 to 31 January 2017).

### 2.3. Aquatic Immersion Test

SS400 and each coated coupon were cut into 2 cm × 1 cm rectangles using a shearing machine (Komatsu, Kanazawa, Japan). Non-coated sides were masked using silicone sealant (Cemedine Co. Ltd., Tokyo, Japan). We used an open laboratory biofilm reactor (LBR) [20] for aquatic biocorrosion accelerated testing (Figure 2). This open LBR was made by Sakai Netsu-Giken, and consisted of a cylindrical column, a water tank, an air fan, and a pump. Each coupon segment was secured to an acrylic holder with an acrylic screw pin. The holder was inserted into an acrylic column that was connected to polyvinylchloride (PVC) pipes at either end. The water tank held approximately 50 L of tap water, and this water was circulated through the open LBR at 6 L/min, at 37 °C for seven days. One patterned holes plate was placed between the water tank and one end of the PVC pipe, which could trap microbes in the atmosphere drawn in by the air fan.

### 2.4. Observation of Morphology and Element Visualization on the Surface of Coupons

The surface morphology of coupons was investigated using field emission scanning electron microscopy (FE-SEM, Hitachi, Tokyo, Japan). In preparation for SEM observation, each coupon was ultrasonically rinsed in acetone to remove contaminants such as corrosion products. To detect specific elements such as calcium and silicon that originate from biofilms [21], iron, silicon, calcium, and zinc or copper were visualized using the mapping function of EDX (Hitachi, Tokyo, Japan) attached to the FE-SEM.

### 2.5. Quantification of Elements on the Surface of Coupons

The amount of each element on the surface of the coupons was evaluated three times using an x-ray fluorescence analyzer (Hitachi, Tokyo, Japan).

### 2.6. DNA Extraction

A PowerSoil^®^ DNA isolation kit (MO Bio Laboratories, Carlsbad, CA, USA) was used for DNA extraction from biofilms formed on the surface of each specimen, whether from the outdoor exposure test or the aquatic biofilm formation test. For the outdoor exposure test, DNA was extracted from deposits on the surface of each outdoor-exposed coupon from two individual areas of each coupon. Biofilm samples from each specimen were scratched off using a sterile spatula, and collected in a PowerSoil^®^ Bead Tube. Sixty µL of C1 solution was added to each tube, tubes were then inverted several times, secured to a bead crusher (TITEC, Koshigaya, Japan), vortexed at 4600 rpm for 1 min, and placed on ice. The vortexing and cooling step was repeated nine more times. Any DNA present in the tubes was then purified according to a previously reported protocol [19]. The concentration of the purified DNA solution was measured using a Qubit fluorometer (Thermo Fisher Scientific Inc., Waltham, MA, USA) and a dsDNA high sensitivity (HS) assay kit (Thermo Fisher Scientific, Waltham, MA, USA).

### 2.7. 16S rRNA Gene-Based Bacterial Community Analysis

The experimental procedure performed according to our previous study in Antibiotics [22].

### 2.8. Raman Spectroscopy Analysis

Before and after exposure testing, the surface of each coupon was observed at 5-fold or 100-fold magnification using the attached microscope of a laser Raman spectroscope (NRS-3100, JASCO Co., Tokyo, Japan). The surfaces were then irradiated with laser light at approximately 649 cm^−1^ (500–2500 cm^−1^) for 0.3 s, and the Raman reflection was measured. About non-exposure tested coupons, five points (in the vicinities of the center and the four corners) were randomly selected for each coupon. Regarding the exposure tested coupons, five spots where deposits were observed were selected for each coupon.

### 2.9. Quantitative Biofilm Formation

After Raman spectroscopic analysis, one section (1 cm × 5 cm) was cut out of each post- exposed coupon. After taking photos, each coupon section was soaked in 0.1% crystal violet solution for 30 min at 25 °C. Treated samples were removed from the solution and rinsed with tap water to remove non-specifically absorbed stain from the surface. Samples were dried for 10 min on a paper towel, then Scotch^®^ mending tape (3M Japan, Tokyo, Japan) was affixed to the polymer-coated side. Thirty minutes later, the tape was removed and affixed to a glass slide. The stained area was measured at five positions using a color reader CR-13 (KONICA MINOLTA, Inc., Tokyo, Japan) from the opposite side to the tape affixed on the glass slide. White paper was used for calibration. Measured data were described using the L*a*b* color system: L* represents lightness (calibration value was 100), a* represents the red/green coordinate (calibration value was zero), and b* represents the yellow/blue coordinate (calibration value was zero). If the color is violet, a* assumes a positive value and b* a negative value. We calculated the three-dimensional vector values (i.e., $\sqrt{{\left(a^{*}\right)}^{2}+{\left(b^{*}\right)}^{2}+{\left(100-L^{*}\right)}^{2}}$) to infer the extent to which the sample formed a biofilm, indicating how a sample formed a biofilm [23].

## 3. Results and Discussion

### 3.1. Outdoor Exposure Tests

All coupons were placed on a roof terrace for two months. The surface color of the SS400 coupons changed from a shiny gray to a drab gray, yellowish gray, and partly rubric brown. The surface color of Cu-coated coupons also changed from shiny bronze to a drab dark brown and rubric brown. The surface color of the Zn-coated coupons changed from a shiny blueish gray to a drab light gray and were partly white (Figure 3). Generally, the color of iron oxides varies according to their oxidative condition. When water and oxygen are present, red-brown-colored rust occurs [24], while zinc oxide is white in color and insoluble in water [25]. Considering this information, the brown-colored area on the SS400 coupons and Cu-coated coupons was probably deposited iron rust such as Fe(OH)_3_. In addition, Cu-coated coupons were covered with larger brown-colored areas than SS400, while there were no brown-colored areas on the surface of Zn-coated coupons. These observations indicated that Cu-coated coupons were more corroded than the SS400 coupons, whereas Zn-coated coupons were not corroded. Furthermore, the surfaces of the Zn-coated coupons were covered with zinc oxide and were therefore in a passive state.

Next, we observed the surface conditions of the coupons both before and after outdoor exposure testing using an optical microscope (Figure 4). For the before outdoor-testing samples, SS400 showed a flat surface; Zn-coated coupons were almost completely covered with the zinc thermal sprayed coating, although some silver color parts of the SS400 basal plate were observed; while the Cu-coated coupons were covered with the copper thermal sprayed coating, but also showed some silver parts as well as red-brown-colored parts, which were probably the SS400 basal plate and iron rust, respectively. The iron rust would be made on the surface of the Cu-coated coupons. As the Cu-coated coupons were very iron corrosive, it would suffer iron corrosion before the test. The surface of a thermal spray coated specimen is irregularly shaped and has gaps because the sprayed metals are aggregated and attached on the surface of a substrate. When moisture exists on the Cu-coated coupon, the coating element (Cu) and the substrate (Fe) are simultaneously immerged in a water solution. Then, Fe becomes an ionized state according to ionization tendency. Ionized irons move to the surface of the Cu-coated coupon through the gaps of spray-coating, and combine with oxygen to form iron oxides there. For the after outdoor-testing samples, the surface of the SS400 coupons showed many small pits (<0.1 mm), cracks, and large holes (3–5 mm). Some parts of the large holes were covered with semi-opaque brown sediments. Zn-coated coupons showed a very rough surface that was white-blue in color, with no silver or brown areas. The Cu-coated coupons also showed a very rough surface, with some parts that were dark brown in color and other parts red-brown. Since pits and holes signify corrosion, and brownish sediments suggest iron rust, the SS400 coupons and Cu-coated coupons incurred iron corrosion, while the Zn-coated coupons incurred zinc corrosion (i.e., the surfaces of Zn-coated coupons were covered with zinc oxide). It appeared that the Cu-coated coupons were more aggressively corroded than the SS400 coupons. Figure 5 shows the SEM images of the surfaces of the SS400, Cu-, and Zn-coated coupons, before and after outdoor exposure testing. Polishing scratches were observed on the SS400 coupons before exposure testing (Figure 5a). Weld splashes of thermally sprayed materials such as copper and zinc were observed in the Cu- and Zn-coated coupons before exposure testing (Figure 5b,c). In addition, many pores were observed on the Cu-coated coupons, which indicated that the density and adhesion of the thermally sprayed layer was low. All coupons were covered with corrosion products following outdoor exposure testing (Figure 5d–f). Iron oxides were detected on the surface of the SS400 and Cu-coated coupons following outdoor exposure testing, while zinc oxides were detected on the surface of the Zn-coated coupons. According to the microscopic images, it was confirmed that the SS400 and Cu-coated coupons underwent iron corrosion, while the Zn-coated coupons underwent zinc corrosion.

Next, sediments on the surfaces of the outdoor-exposed coupons were identified as biofilms by Raman spectroscopic analysis, with the amount of biofilm then quantified using crystal violet staining. In addition, DNA extracted from the sediments was used for bacterial microbiota analysis.

A biofilm consists of water (~80%), microbes, and exstracellular polymeric substances (EPSs). EPSs are a mixture of polysaccharides, extracellular DNA (eDNA), lipids, and proteins [26], and they persist even if biofilm-forming bacteria disappear from a mature biofilm. Since many EPS components can be detected by Raman spectroscopy, several Raman peaks derived from lipids, nucleic acids, proteins, and polysaccharides have been reported [27,28,29,30,31,32,33,34,35]. Since most reported Raman peaks involving biological components have been detected at 500–1800 cm^−1^, there is unfortunately relatively little information about biological components in the range of 1900–2300 cm^−1^. Figure 6 shows the results of Raman spectroscopic analysis of outdoor-exposed coupons. All coupons were found to have several Raman peaks derived from EPSs. For SS400 coupons, amide III-derived peaks (1199–1347 cm^−1^) [34] had the strongest relative intensity. For Zn-coated coupons, peaks in the >1700 cm^−1^ region were stronger than those in the <1700 cm^−1^ region, with the >1700 cm^−1^ region including carbonyl compound-related peaks (1706–1918 cm^−1^) [34] and OH–NH–CH stretching vibration peaks associated with nucleic acids (2313–2500 cm^−1^ in SS400 coupon, 2290–2498 cm^−1^ in Cu-coated coupon, 2292–2500 cm^−1^ in Zn-coated coupon) [32]. For the Cu-coated coupons, tyrosine-derived peaks (643–696 cm^−1^) [29] were the strongest. According to these results, biofilms existed on the surfaces of the SS400, Zn-, and Cu-coated coupons following the outdoor exposure testing, however, the EPS contents probably differed among them.

Crystal violet staining is one of the methods used for biofilm biomass quantification [36]. The density of crystal violet staining was measured using a colorimeter, then the modulus of the combination of chromaticity value and brightness ($\sqrt{{\left(a^{*}\right)}^{2}+{\left(b^{*}\right)}^{2}+{\left(100-L^{*}\right)}^{2}}$) was calculated. When a biofilm is stained using crystal violet, the color changes to violet and the transparence is lower (i.e., chromaticity of a* is positive value, that of b* is negative value, and brightness, L*, is lower than 100). If a biofilm becomes thicker (mature), $\sqrt{{\left(a^{*}\right)}^{2}+{\left(b^{*}\right)}^{2}+{\left(100-L^{*}\right)}^{2}}$ is larger. Therefore, the amounts of biofilms can be quantified using the value of $\sqrt{{\left(a^{*}\right)}^{2}+{\left(b^{*}\right)}^{2}+{\left(100-L^{*}\right)}^{2}}$. The value for the Zn-coated coupons was significantly smaller than that of the SS400 coupons, the value for the Cu-coated coupons, however, was similar to that of the SS400 coupons (Figure 7). This result indicated that the Zn-coated coupons had a lower biofilm formation than the SS400 and Cu-coated coupons. Kanematsu et al. reported that *Pseudomonas aeruginosa* and *Pseudoalteromonas carageenavara* formed biofilms on the surface of SS400 much more than that of the other metal plated steels such as tin-, copper-, and zinc-plated ones, and iron (ion) could pull these bacteria better than other metals including tin, copper, and zinc [37]. In this study, iron oxide was detected on the surface of the coupons and that of the Cu-coated coupons after the outdoor exposure test (Figure 5), on the other hand, zinc oxide was detected on the surface of Zn-coated coupons. Therefore, it was probably that the surface of SS400 coupons and that of Cu-coated coupons were rich in iron (ions), but that of Zn-coated coupons was poor in iron, resulting in the difference of biofilm formation among these coupons.

16S rRNA gene analysis was performed to reveal which bacterial groups were related to biofilm formation and microbially influenced corrosion (MIC). Some species are known to be MIC-related bacteria. For example, *Pseudomonas aeruginosa* can induce MIC on steel and stainless steel under aerobic conditions [10,38,39]; sulfate-reducing bacteria (SRB) such as *Desulphovibrio vulgarius* produce H_2_S, which triggers iron ionization under anaerobic conditions [40,41]; sulfur-oxidizing bacteria such as *Acidithiobacillus thiooxidans* (old name: *Thiobacillus thiooxidans*) produce sulfuric acid which increases acidity and induces iron oxidization [6,42,43]; and iron-oxidizing bacteria such as *Callionella* and *Leptothrix* oxidize iron [6,44]. *Desulphovibrio*, *Acidithiobacillus, Pseudomonas*, *Callionella*, and *Leptothrix* belong to the orders Desulfovibrionales, Acidithiobacillales, Pseudomonadales, Nitrosomonadales, and Burkholderiales, respectively. Figure 8 shows the results of bacterial microbiome analysis related to outdoor-exposed coupons. The most abundant bacterial orders were Actinomycetales (25%, #2 in Figure 8) and Burkholderiales (25%, #18 in Figure 8) in SS400–1; Actinomycetales (26%) in SS400–2; Bacillales (32%, #11 in Figure 8) and Burkholderiales (32%) in Cu-coated–1; Pseudomonadales (49%) in Cu-coated–2; Burkholderiales (29%) in Zn-coated–1; and Flavobacteriales (18%, #4 in Figure 8) and Pseudomonadales (17%, #21 in Figure 8) in Zn-coated–2. In these three kinds of coupons, bacterial order occupancies were very different between one area and the other, even on the same coupon. However, some common bacterial orders, namely Actinomycetales, Bacillales, and Pseudomonadales were detected on all coupons, although the degrees of occupancy were different among them. Actinomycetales and Pseudomonadales are dominant bacterial orders in soil communities [45,46]. From this, it can be inferred that major bacterial groups detected from the biofilms came from soil carried by the wind. In addition, Bacillales can survive starvation by forming dormant and resistant spores [47], therefore, Bacillales probably survived as spores in the biofilms of all samples. The top three most abundant genera in each sample are summarized in Table 1. The second-most abundant genus, *Pseudomonas*, seen on Zn-coated-1, was the only MIC-related bacteria, however, most bacteria are reported as being able to form biofilms [48,49,50,51,52]. These findings imply that MIC-related bacteria existed in very low quantities in the biofilms formed on these coupons and were rarely involved in the corrosion. Additionally, the abundant genera of biofilm-forming bacteria were very different in samples with the same coated coupons, indicating that biofilms were partly removed from the surface of the coupons when corrosive products (rusts) were eroded by wind or rain during the exposure period.

In the outdoor exposure test, Cu-coated coupon was the most iron corroded followed by SS400 coupon, while Zn-coated coupon was barely iron corroded at all because of the sacrificial corrosion of zinc. The amount of biofilm on Zn-coated coupons was significantly lower than that on the SS400 coupons. These results implied that the tendency for iron corrosion was negatively correlated with that of biofilm formation. Meanwhile, the abundance of MIC-related bacteria was unrelated to the tendency for corrosion. Considering the little relationship between biofilm formation and corrosion, chemical corrosion (i.e., non-MIC) would dominantly progress than MIC on the surface of specimens under outdoor-exposed condition, and biofilms would play an important role in triggering corrosion because biofilms can store water, a crucial factor for corrosion in steel [53]. In addition, the outdoor exposure testing position was located 2 km from Ise Bay, so the air would be rich in sodium chloride, an accelerating factor for corrosion in atmospheric conditions (ionic conditions) [54]. Generally, the initiation of corrosion needs ionization, which tends to occur in aquatic conditions. Biofilms can retain moisture from the air on the surface of the coupons, which makes iron ionization easier. Based on these factors, we inferred a corrosion process under outdoor exposure conditions as follows (Figure 9). First, environmental bacteria were carried to the surface of SS400, Zn-, and Cu-coated coupons by the wind (Step 1). Next, these organisms attached to the surfaces, proliferated, and produced EPSs, resulting in biofilms (Step 2). The process of corrosion is the result of electrochemical reactions i.e., an iron atom changes its oxidation state, then ionized iron is released from the solid surface and combines with H_2_O to form iron oxide (iron rust) [55,56]. The dissolution of iron atoms into iron ions would depend on the porosity of the surface films, and the potential difference between the substrate metal (iron) and the film constituent (metal used for thermal spray coating). Since thermal spray coated films are generally porous, the results differ from the kinds of coated metal to metal. The initiation of corrosion needs water and oxygen. Biofilms act as water storage locations where metals are dissolved and transformed, resulting in mineral formation [6], according to the ionization tendency of zinc, iron, and copper. Zinc is more easily ionized than iron, therefore it occurs as insoluble zinc oxide or zinc hydroxide, which will cover the surface of the Zn-coated coupons, terminating any further iron ionization. Meanwhile, iron is more easily ionized than copper, therefore the surface of the Cu-coated coupon was more corroded than that of SS400 coupon (Step 3). Indeed, the surfaces of the Cu-coated coupon showed more rust than those of SS400 coupon. At the same time, sodium chloride in the atmosphere dissolved in the biofilms and accelerated electrochemical corrosion (Step 4).

### 3.2. Aquatic Immersion Tests

All specimens were set in the open LBR, where tap water was circulated for seven days. The surface of the SS400 coupons had some brown-colored spots (iron rust), the surface of the Cu-coated coupons had more brown-colored spots than that of SS400 coupons, while the surface of the Zn-coated coupons was mostly covered in green or brown-colored sediments (Figure 10). In the optical microscopy images obtained following the LBR immersion test, black-colored concentric holes (about 2.5 mm diameter) were observed on the surface of the SS400 coupons, dark brown-colored pockets (>3 mm-long axes) were observed on the surface of the Cu-coated coupons, whereas most parts of the surfaces of the Zn-coated coupons were covered with brown or green-brown-colored sediments, while the remainder on the Zn-coated coupons were white-blue in color (Figure 11). Brown-colored holes/pockets seemed to be corroded, whereas green-brown-colored sediments seemed to be cyanobacteria-rich biofilms.

SEM analysis showed that areas that bulged outward were observed on the surfaces of SS400, Cu- and Zn-coated coupons post-water-immersion testing (Figure 12). Since SS400, a carbon steel, consists of carbon and steel, these bulging sediments were obviously iron oxides (i.e., corroded iron products). In order to confirm whether or not the bulging areas arose from iron oxides or other compounds, an element mapping was conducted. On the surface of the Cu-coated coupons, areas with iron corresponded to areas with oxygen and silicon (Figure 13). Meanwhile, on the surface of the Zn-coated coupons, zinc areas corresponded to areas with oxygen and silicon (Figure 14). These results indicate that bulging areas of the surface of the Cu-coated and Zn-coated coupons were iron oxides and zinc oxides, respectively. Interestingly, these oxides were corrosion products. According to ionization tendency, zinc is more easily ionized than iron, while copper is barely more ionized than iron. Therefore, Zn-coated coupons and Cu-coated coupons will cause mainly zinc corrosion and iron corrosion, respectively. Indeed, detected corrosion products reflected the differences in ionization tendency between zinc and iron, or copper and iron. In addition, Kuroda et al. reported that silicon was detected with calcium in biofilms in cooling water systems [21]. Considering that silicon and calcium are derived from tap water, biofilms would be formed on the corroded sites of the Cu-coated coupons, Zn- coated coupons, and SS400 coupons by tap water and air-trapped bacteria.

Next, biofilm formation was measured both quantitatively and qualitatively. Raman analysis showed that many peaks assigned to EPSs were detected on the surface of all coupons after water-immersion testing (Figure 15). These results indicated that biofilms formed on the surfaces of all coupons. Additionally, all coupons showed that relative intensities more than 2000 cm^−1^ were higher than those for less than 2000 cm^−1^, although the pattern of Raman peaks varied among samples. Considering the results of the Raman peaks, it remains unclear as to how the biofilms formed on the SS400, Cu-, or Zn-coated coupons were similar (or different), because there is very little information about Raman peaks in the more than 2000 cm^−1^ region that have been assigned in EPSs.

Compared with the amount of biofilm that formed on the SS400 coupons, significantly more biofilms formed on the Cu-coated coupons, while the amount of biofilm that formed on the Zn-coated coupons was significantly lower (Figure 16). Ranking the three specimens in descending order of biofilm quantity, the order (Cu-coated coupons > SS400 coupons > Zn-coated coupons) corresponds exactly to their corrosion tendency. This indicates that the larger the biofilms that form on the surface of coupons in an aquatic environment, the more corrosion proceeds.

Bacterial microbiome analysis showed that the occupancy pattern of taxonomic orders was similar in duplicate microbiomes of Zn-coated coupons and Cu-coated coupons, but not for the SS400 coupons (Figure 17). Rhodocyclales (#19 in Figure 17) was the most dominant in all samples; its occupancy on the Cu-coated coupons was just under 40%, while for the SS400 and Zn-coated coupons, it was more than 50%. However, the main dominant genera were different among them (Table 2): for the SS400 coupons, they were *Methyloversatilis* and an untitled genus, for Zn-coated coupons it was *Methyloversatilis*, but for Cu-coated coupons it was an untitled genus. Additionally, Sphingomonadales and Stramenopiles were characteristic orders found on the Cu- and Zn-coated coupons, respectively. The main genus of Sphingomonadales on Cu-coated coupons was *Sphingomonas* (Table 3), while for Stramenopiles on Zn-coated coupons, it was an unclassified new genus. Stramenopiles include ecologically important algae such as diatoms and kelp as well as heterotrophic and parasitic bacteria [57]. Algae are usually green or brown because of their chlorophylls, which will influence the color of biofilms. Stramenopiles was an abundant order (about 10%) on Zn-coated coupons, which is probably the reason why the biofilm on the Zn-coated coupons was green-brown in color (Figure 10 and Figure 11). Many members of the Rhodocyclales have the capability to remove anthropogenic compounds from the environment and biological systems (i.e., to utilize various carbon compounds) [58]. *Methyloversatilis* comprises three species that can utilize single carbon compounds such as methanol and methylamine, as a sole source of energy [59,60,61]. This implies that biofilm-forming bacteria actively participate in the biodegradation of chemical compounds derived from metal coupons and/or the circulating LBR water. Moreover, *Sphingomonas* is known to be a nuisance in copper pipes used in drinking water distribution systems because it can accumulate copper ions in its cell wall and use these copper ions as binders for facilitating anodic reactions in the MIC of copper [62]. *Sphingomonas paucimobilis* is also reported to be a major biofilm producer [63], but copper can generally kill these bacteria by contact killing [64]. According to these bacterial features, the following story was inferred: a small amount of copper ions was eluted from the surface of Cu-coated coupons where *Sphingomonas* was preferentially drawn, forming a biofilm. Subsequently, a member of Rhodocyclales was recruited and became dominant due to iron ions eluted from the surface of the Cu-coated coupons, and attached to the biofilm, resulting in the progression of biofilm formation. On the surface of the SS400 coupons, iron ions were eluted and attracted many kinds of bacteria including the members of Rhodocyclales. Rhodocyclales probably proliferated dominantly because they could use various carbon compounds, then formed biofilms. On the surface of the Zn-coated coupons, zinc ions were eluted at the initial stage of this experience and also recruited some bacteria because zinc is known to be an essential element in many bacteria [65]. Next, the zinc ions bound to oxygen to make zinc oxides that inhibited the elution of zinc ions, which could limit the biofilm formation. Some bacteria are known as MIC-related bacteria, as listed in the previous section, for example, *Pseudomonas*, SRB such as *Desulphovibrio*, and iron-oxidizing bacteria such as *Acidthiobacillus* (*Thiobacillus*) and *Gallionella*. As shown in Table 4, few MIC-related bacteria were found on any of the LBR-immersed coupons. This result indicates that MIC-related bacteria were not involved in the biofilm formation or corrosion of the coupons in the LBR environment.

### 3.3. Comparison of Corrosion in the Outdoor Exposure Test with that of the Water-Immersion Test

In both the outdoor exposure test and the water-immersion test, the Cu-coated coupons were the most intensively corroded, while the Zn-coated coupons were almost entirely covered with zinc oxide, and not iron oxide. This iron corrosion tendency of the Cu-coated coupon, Zn-coated coupon, and SS400 (Cu-coated coupon > SS400 coupon > Zn-coated coupon) was negatively correlated with the ionization tendencies of zinc, iron, and copper (zinc > iron > copper). Biofilms were detected on all coupons following exposure outdoors or in water, with the Zn-coated coupon found to have the smallest amount of biofilm in both trials. Based on these results, it seems likely that biofilm formation was initially involved in corrosion in both the outdoor and the water exposure tests. However, electrochemical corrosion was the dominant process in the outdoor exposure tests, while MIC was the dominant process in the water exposure tests. Indeed, a mixture of silicon and calcium was detected on the surface of the water-exposed SS400, Cu-, and Zn-coated coupons (Table 5). Based on a previous report that calcium and silicon are constituent elements of biofilms [21], biofilms formed well on the surface of the coupons immersed in water. After the exposure tests, the presence of coating elements (copper and zinc) decreased, while that of iron simultaneously increased. The range of decreases and increases was higher for the water exposure tests compared with the outdoor exposure tests. These results showed that an aquatic environment accelerated MIC more than an atmospheric environment.

## 4. Conclusions

In this study, we investigated the effect of zinc thermal spray coated carbon steel (Zn-coated) and copper thermal spray coated carbon steel (Cu-coated) on iron corrosion in air or an aquatic environmental condition. We also explored which factor(s) such as biofilm formation (MIC) and ionization tendency (electrochemical corrosion) worked dominantly in iron corrosion. We expected that the Zn-coated and Cu-coated could inhibit iron corrosion due to the sacrificial corrosion of zinc and contact killing of biofilm-forming bacteria, respectively. In both air and water environments, the Zn-coated inhibited iron corrosion, but the Cu-coated accelerated iron corrosion, which was negatively correlated to ionization tendency (i.e., Zn > Fe > Cu). The dominant factor of iron corrosion will be electrochemical corrosion in the air environment and MIC in the water environment, however, it is probable that biofilm formation plays an important role in iron corrosion. In an air environment, biofilms can store water (moisture) that makes galvanic cells elute metallic ions according to ionization tendency. In a water environment, biofilms can accelerate iron corrosion caused by bacterial metabolites [66]. MIC-related bacteria have been found in specific environments such as oil tanks and water systems of nuclear power plants. In this study, MIC-related bacteria were barely detected in the biofilms formed on the surface of the Zn-coated, Cu-coated, and carbon steel. This implies that biofilm formation is an essential factor for iron corrosion, but MIC-related bacteria are not always necessary for it. Additionally, iron ions can predominantly attract bacteria more than zinc ions and copper ions, therefore, inhibiting the elution of iron (ions) will be an effective approach to regulate biofilm formation as well as iron corrosion.

## Figures and Tables

**Figure 1 materials-13-00923-f001:**
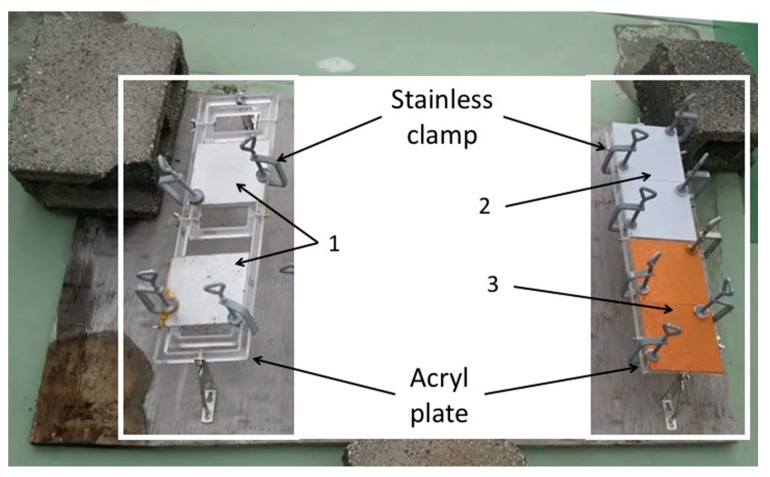
The apparatus for the outdoor exposure test. 1: SS400; 2: Zn-coated; 3: Cu-coated.

**Figure 2 materials-13-00923-f002:**
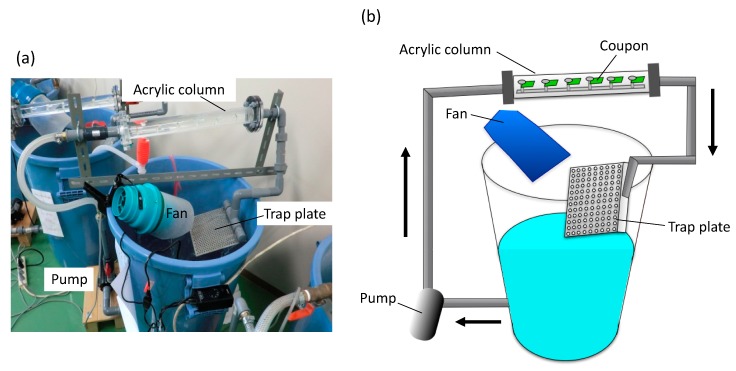
The laboratory biofilm reactor (LBR). (**a**) Photo image of the LBR. (**b**) The outline of the LBR. Arrows indicate water flow.

**Figure 3 materials-13-00923-f003:**
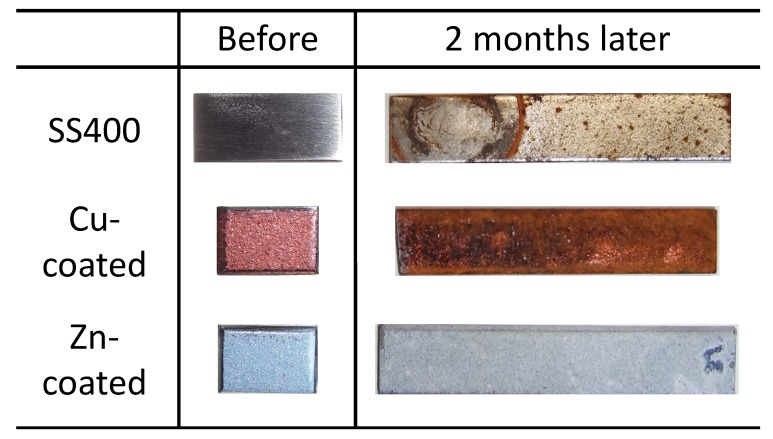
Digital photo images of the outdoor-exposed specimens.

**Figure 4 materials-13-00923-f004:**
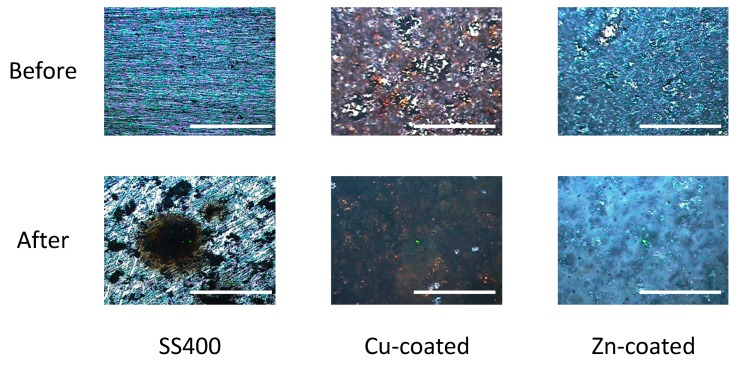
Optical microscopy images of the surface of the specimens before and after outdoor exposure tests. Each white bar represents 5 mm. A green center point is a laser-irradiated spot.

**Figure 5 materials-13-00923-f005:**
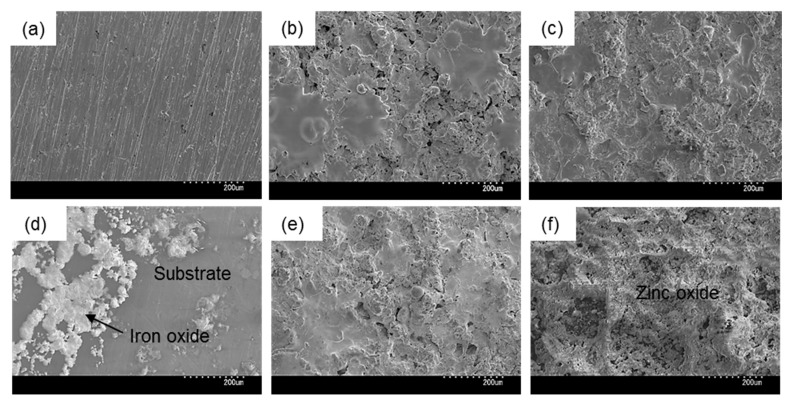
SEM images of the surface of coupons before (**a–c**) and after (**d–f**) outdoor exposure. (**a**) and (**d**) SS400; (**b**) and (**e**) Cu-coated; (**c**) and (**f**) Zn-coated.

**Figure 6 materials-13-00923-f006:**
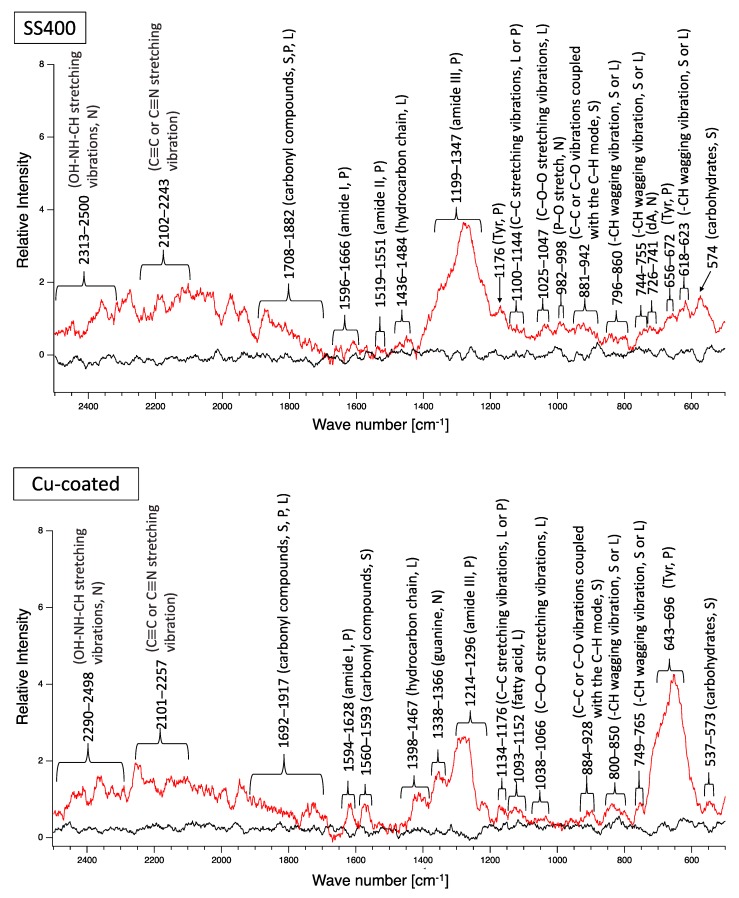
Raman peaks of sediments on the surface of the coupons following outdoor exposure testing (red line). Black lines show the Raman peaks of the surface of the specimens before the test. Detected Raman peaks after the test were assigned to related chemical bonds of compounds according to information in references [27,28,29,30,31,32,33,34,35]. N: nucleic acids; L: lipids; P: proteins; S: polysaccharides.

**Figure 7 materials-13-00923-f007:**
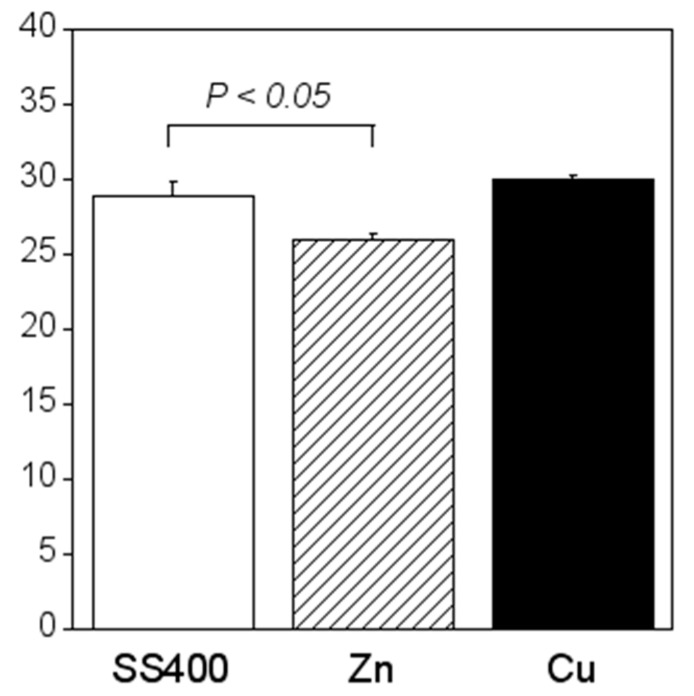
Biofilm quantification of outdoor-exposed coupons. Each column shows the mean of the modulus of $\sqrt{{\left(a^{*}\right)}^{2}+{\left(b^{*}\right)}^{2}+{\left(100-L^{*}\right)}^{2}}$ (n = 5). Zn and Cu mean Zn-coated coupons and Cu-coated coupons, respectively. Error bars show the standard deviation. Student’s t-test was performed between the SS400 coupons and Zn-coated coupons or SS400 coupons and Cu-coated coupons. P means the p-value.

**Figure 8 materials-13-00923-f008:**
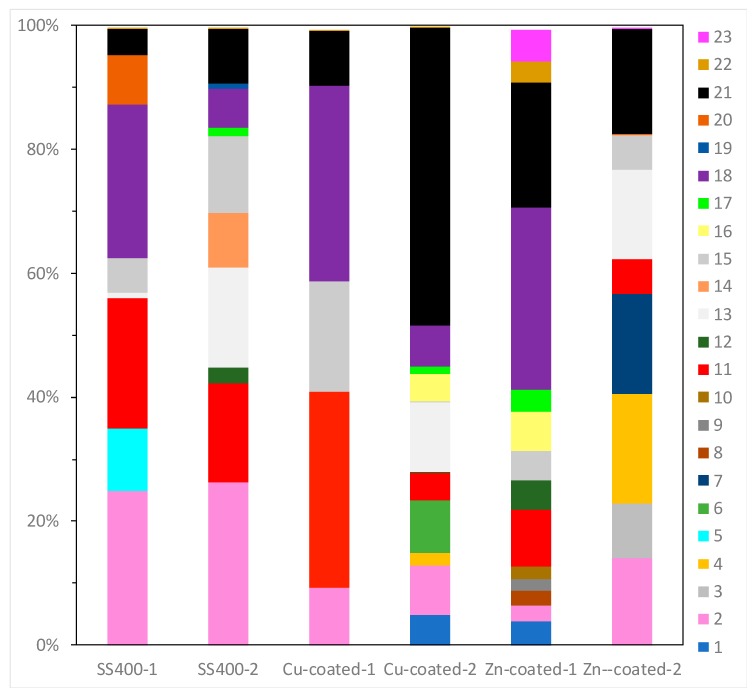
OTU abundance percentages of outdoor-exposed coupons. Bacterial orders present at <1.0% and unassigned OTUs are excluded from each column. (1) Unknown member of Acidobacteria-5; (2) Actinomycetales; (3) Gaiellales; (4) Flavobacteriales; (5) Sphingobacteriales; (6) Saprospirales; (7) G30-KF-AS9; (8) Streptophyta; (9) Stigonematales; (10) Chroococcales; (11) Bacillales; (12) Lactobacillales; (13) Clostridiales; (14) Pirellulales; (15) Rhizobiales; (16) Rhodospirillales; (17) Sphingomonadales; (18) Burkholderiales; (19) Campylobacterales; (20) Legionellales; (21) Pseudomonadales; (22) Xanthomonadales; (23) Chthoniobacterales.

**Figure 9 materials-13-00923-f009:**
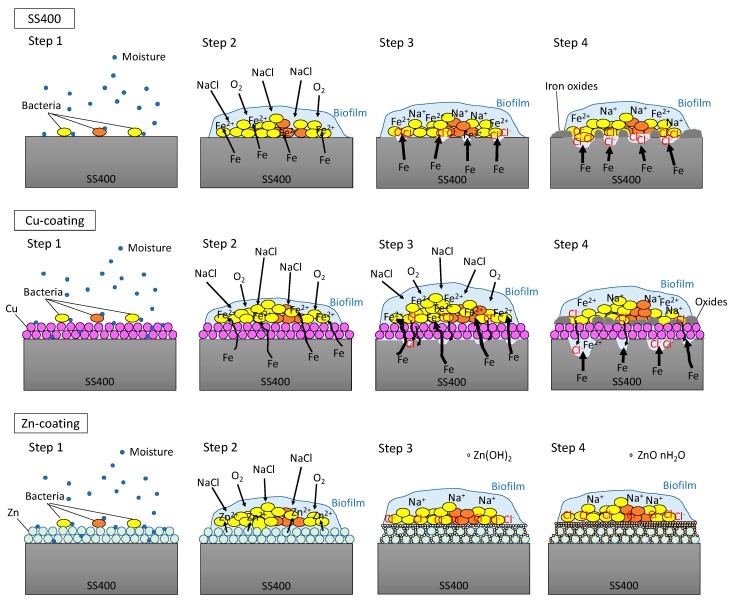
Predictive corrosion progression of the SS400 coupon, Zn-coated coupon, and Cu-coated coupon under outdoor conditions.

**Figure 10 materials-13-00923-f010:**
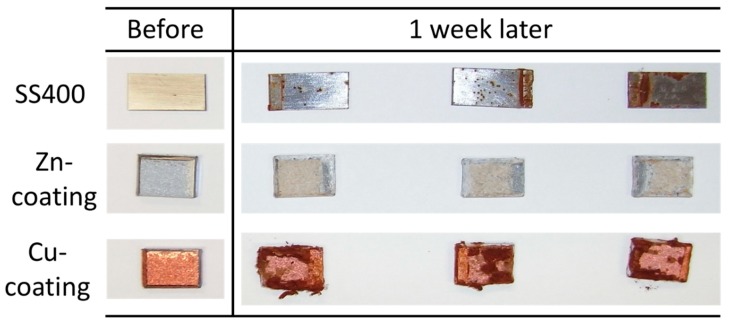
Digital photometric images of the surface of LBR-immerged coupons.

**Figure 11 materials-13-00923-f011:**
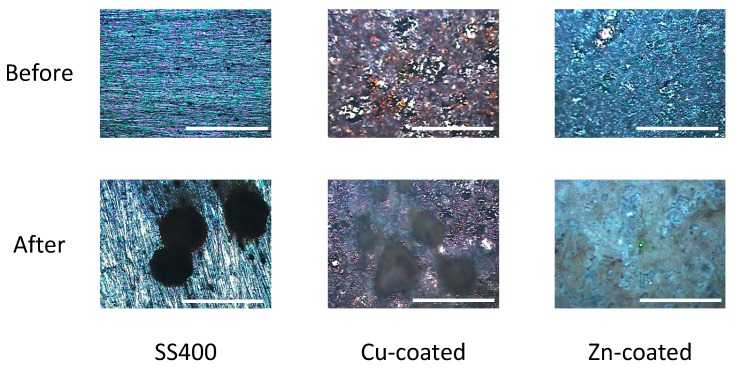
Optical microscopy images of the surface of the LBR-immerged coupons. Each white bar represents 5 mm.

**Figure 12 materials-13-00923-f012:**
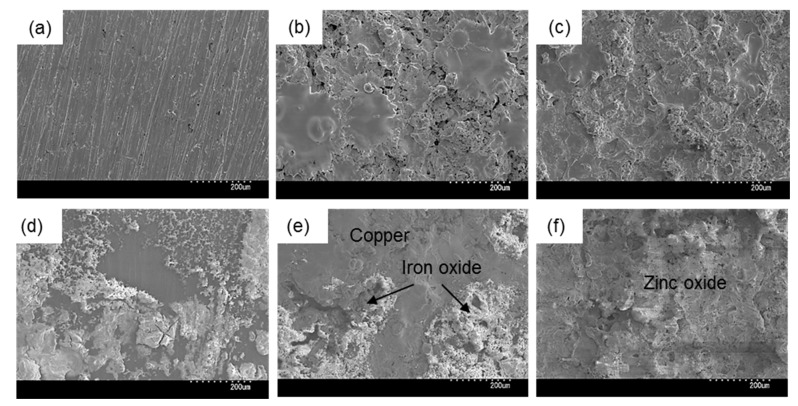
SEM images of the surface of coupons before (**a–c**) and after (**d–f**) LBR immersion testing. **a** and **d**: SS400; **b** and **e**: Cu-coated; **c** and **f**: Zn-coated.

**Figure 13 materials-13-00923-f013:**
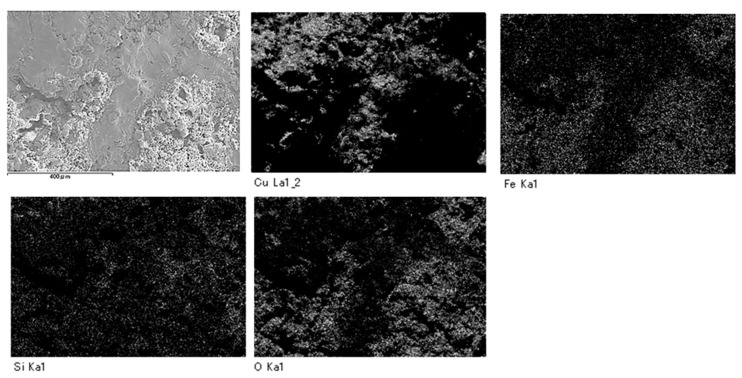
Element mapping images of Cu-coated coupons after LBR-immersion testing.

**Figure 14 materials-13-00923-f014:**
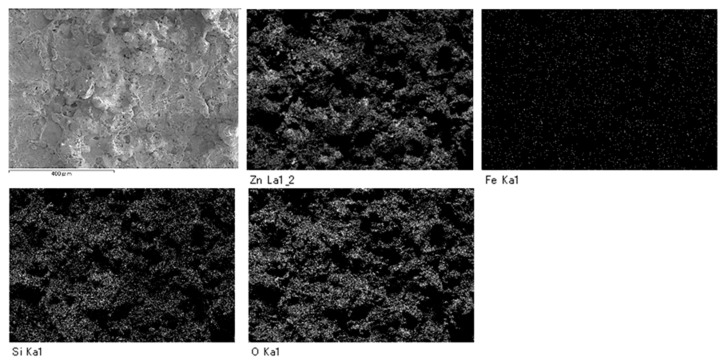
Element mapping images of the Zn-coated coupons after LBR-immersion testing exposure testing in water.

**Figure 15 materials-13-00923-f015:**
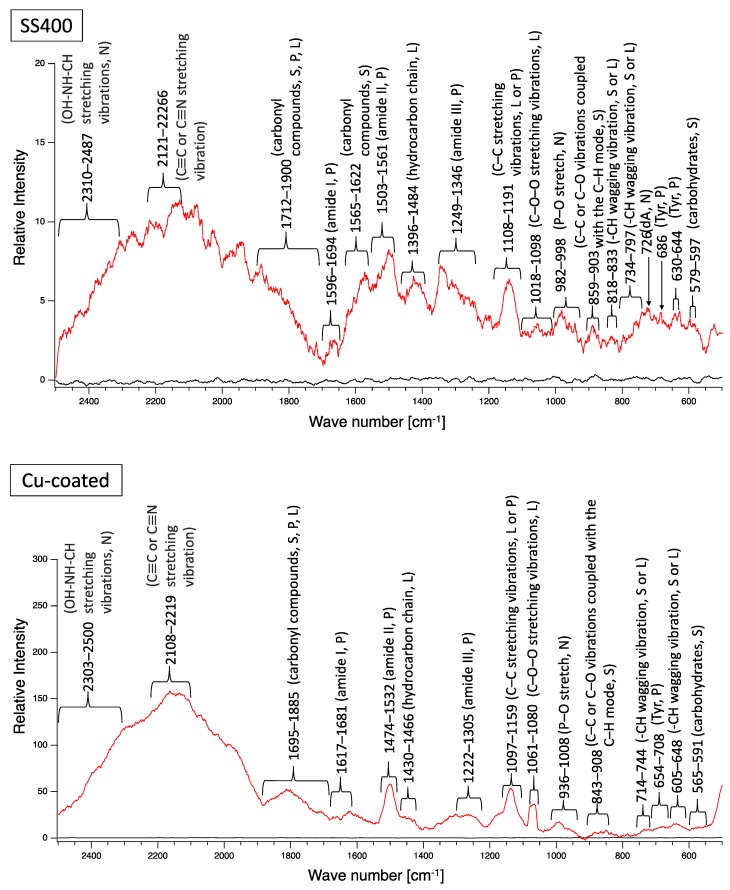
Raman peaks of sediments on the surface of the coupons after LBR immersion testing (red lines). Black lines show the Raman peaks of the surface of the specimens before the test. Detected Raman peaks after the test were assigned to related chemical bonds or compounds according to information in references [27,28,29,30,31,32,33,34,35]. N: nucleic acids; L: lipids; P: proteins; S: polysaccharides.

**Figure 16 materials-13-00923-f016:**
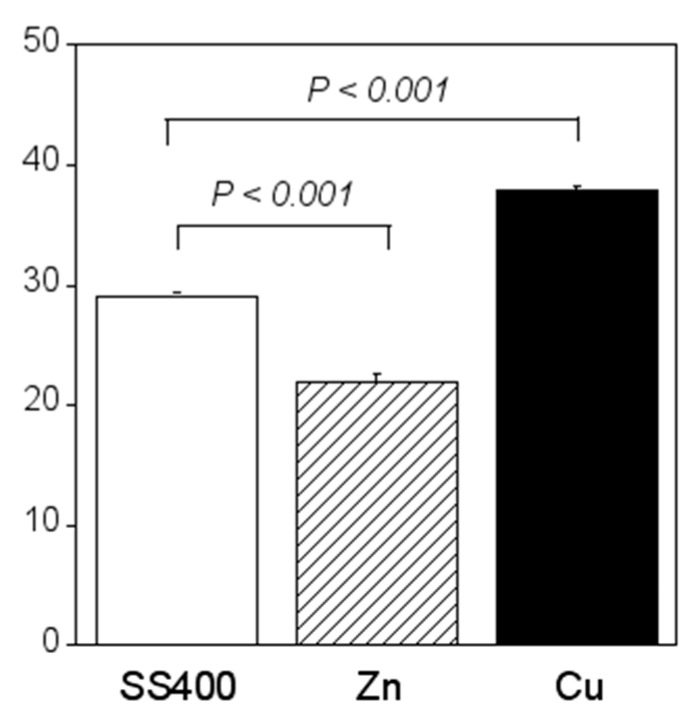
Biofilm quantification of the LBR-immersed coupons. Each column shows the mean of modulus of $\sqrt{{\left(a^{*}\right)}^{2}+{\left(b^{*}\right)}^{2}+{\left(100-L^{*}\right)}^{2}}$ (n = 5). Zn and Cu mean the Zn-coated coupons and Cu-coated coupons, respectively. Error bars show the standard deviation. Student’s t-test was performed between the SS400 coupons and Zn-coated coupons or SS400 coupons and Cu-coated coupons. P means the p-value.

**Figure 17 materials-13-00923-f017:**
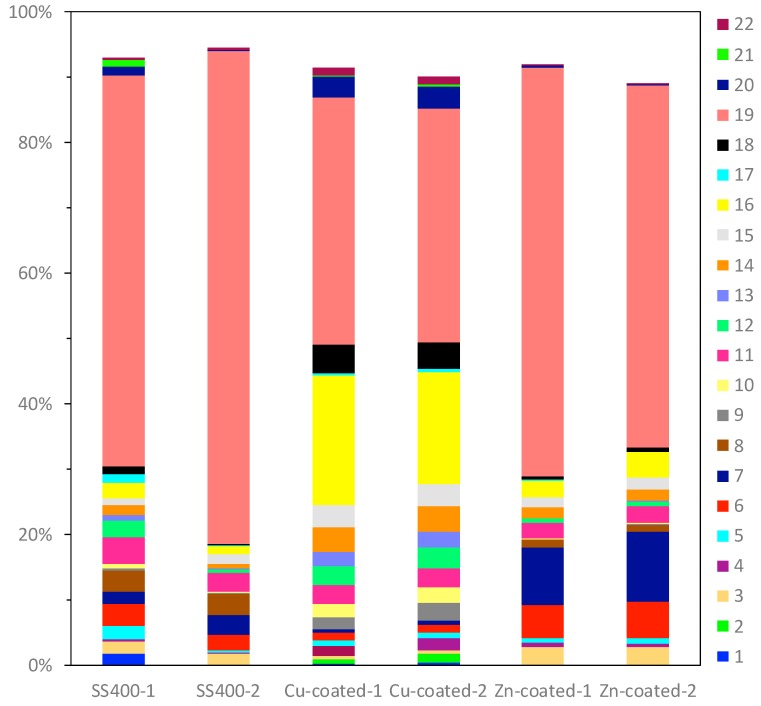
The OTU abundance percentages on water-immersed coupons. Bacterial orders present at <1.0% and unassigned OTUs are excluded from each column. (1) Solibacterales; (2) PK29; (3) Actinomycetales; (4) Cytophagales; (5) Saprospirales; (6) MLE1-12; (7) Stramenopiles; (8) Clostridiales; (9) Gemmatales; (10) Pirellulales; (11) Caulobacterales; (12) Rhizobiales; (13) Rhodobacterales; (14) Rhodospirillales; (15) Rickettsiales; (16) Sphingomonadales; (17) Unknown member of Betaproteobacteria; (18) Burkholderiales; (19) Rhodocyclales; (20) Myxococcales; (21) Unknown member of VHS-B5-50; (22) Unknown member of SJA-4.

**Table 1 materials-13-00923-t001:** Abundant genera of the outdoor-exposed coupons. Each column indicates the assigned genus name and percentage occupancy in parentheses. The numbers in brackets indicate the occupancy of each genus (%).

Rank	SS400-1	SS400-2	Cu-Coated-1	Cu-Coated-2	Zn-Coated-1	Zn-Coated-2
1^st^	*Bacillus* (19.8)	*Staphylococcus* (15.9)	New someone of Oxalobacteraceae (18.7)	*Acinetobacter* (47.9)	New someone of Oxalobacteraceae (19.5)	*Acinetobacter* (17.0)
2^nd^	New someone of Oxalobacteraceae (14.8)	*Peptoniphilus* (9.1)	*Rhodoplanes* (17.8)	New someone of Chitinophagaceae (8.4)	*Pseudomonas* (19.0)	New someone of JG30-KF-AS9 (order) (16.1)
3^rd^	Someone of Intrasporangiaceae (12.9)	*Corynebacterium* (8.9)	*Bacillus* (16.7)	New someone of Ruminococcaceae (7.8)	*Janthinobacterium* (6.7)	*Coprococcus* (14.3)

**Table 2 materials-13-00923-t002:** The OTU percentages of order Rhodocyclales on post LBR-immersed coupons. All assigned genera are from the family Rhodocyclaceae.

Genus	SS00-1	SS400-2	Cu-Coated-1	Cu-Coated-2	Zn-Coated-1	Zn-Coated-2
*Azoarcus*	0.15	0.09	0	0	0.15	0.19
*Azospira*	0.02	0.05	0.02	0.02	0.01	0.01
*Dechloromonas*	1.24	2.17	1.13	1.06	0.67	0.74
*Methyloversatilis*	39.01	22.25	0.23	0.27	55.31	48.43
*Rhodocyclus*	0.03	0.01	0	0	0.03	0.03
untitled	16.92	48.76	36.27	34.25	3.74	3.40
new genus	1.57	1.22	0.18	0.12	1.68	1.61

**Table 3 materials-13-00923-t003:** The OTU percentages of order Sphingomonadales on post LBR-immersed coupons.

Family	Genus	SS400-1	SS400-2	Cu-coated-1	Cu-coated-2	Zn-coated-1	Zn-coated-2
Erythrobacteraceae	*Azoarcus*	0	0	0.04	0.03	0	0
Sphingomonadaceae	*Novosphingobium*	0.07	0.02	0.13	0.15	0.13	0.22
	*Sphingobium*	0.18	0.10	0.11	0.12	1.17	2.31
	*Sphingomonas*	2.08	1.08	19.32	16.58	1.07	1.22
	untitled	0.01	0	0.02	0.03	0.01	0.01
	new genus	0.05	0.02	0.17	0.14	0.02	0.02

**Table 4 materials-13-00923-t004:** The OTU percentages of MIC-related bacteria on post water-immersed coupons.

Family	Genus	SS400-1	SS400-2	Cu-coated-1	Cu-coated-2	Zn-coated-1	Zn-coated-2
Pseudomonadaceae	*Pseudomonas*	0.01	0	0.01	0	0.01	0
Hydrogenophilaceae	*Thiobacillus*	0	0	0	0	0	0
Gallionellaceae	*Gallionella*	0	0	0	0	0	0
Desulfarculaceae	new genus	0	0	0	0	0	0
Desulfobulbaceae	new genus	0	0	0	0	0	0
Desulfuromonadaceae	untitled	0	0	0	0	0	0
Geobacteraceae	new genus	0	0	0	0	0	0

**Table 5 materials-13-00923-t005:** Main components of the surface of each coupon before and after the exposure tests (mass %).

Coupon	Test Type	Element				
		Fe	Cu	Zn	Ca	Si
SS400	before	100	0	0	0	0
	outdoor exposure	99.98	0	0	0.02	0
	water exposure	98.41	0	0	0.36	1.23
Cu-coated	before	0.68	99.01	0	0	0.31
	outdoor exposure	6.27	93.73	0	0	0
	water exposure	11.34	77.95	0	8.46	2.25
Zn-coated	before	0.09	0	99.52	0	0.39
	outdoor exposure	0.13	0	99.17	0	0.70
	water exposure	0.33	0	90.23	2.84	6.60

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
