# Peer review of "Biofilm Formation Plays a Crucial Rule in the Initial Step of Carbon Steel Corrosion in Air and Water Environments"

_materials, 2020, doi:10.3390/ma13040923_

Round 1
Reviewer 1 Report
Dear Authors please find attached the PDF file with the review.

Author Response
Dear Reviewer,
Thank you very much for your productive comments. We revised the manuscript according to your comments. We indicated the rewritten parts in red color.
In the original paper, we refer to SS400 carbon steel plate, zinc thermal spray-coated SS400 carbon steel plate, and copper thermal spray-coated SS400 carbon steel plate, as SS400, Zn-coating, and Cu-coating, respectively. However, we sometimes used “Cu-coated coupons” and “Zn-coated coupons” there. These mixing expressions were very confusing. Therefore, we conformed Cu-coating” and “Zn-coating” to “Cu-coated” and “Zn-coated” , respectively, including figures and tables.
We also transferred the following sentences to after “Cu-coated coupons was Sphingomonas (Table 3), while for Stramenopiles on Zn-coated coupons it was an unclassified new genus.” (page 18, lines 21-25) because we thought the position much better than before to understand our idea easily.
Additionally, we added corrosion story in a water environment about Zn-coated coupons and SS400 coupons at page 19, lines 9-15.
We hope it will become better and clearer scientific article than the original version.
<Questions and answers>
Q1) The abstract should be shortened. The highlighted part should be omitted.
We deleted following sentences from the abstract.
Zinc coating is known as an efficient method to prevent carbon steel corrosion by sacrificial protection in both air- and water-exposed conditions. Biofilm formation is a well-known nuisance for causing carbon steel corrosion, i.e., microbially influenced corrosion (MIC). Many studies have reported a relationship between carbon steel and MIC but most of them have focused on specific conditions, such as anaerobic environments, in crude and marine conditions. In addition, certain bacteria are sometimes used for biofilm formation.
Q2) Should add the references about electrical corrosion and MIC.
We listed related references as follows.
Khoshnaw, F.; Gubner, F. (Edi.) Corrosion Atlas Case Studies 2019 Edition. Amsterdam and Oxford, Elsevier (2019). E. Huttunen-Saarivirta; P. Rajala; M. Marja-aho; J. Maukonen; E. Sohlberg; L. Carpén; Ennoblement, corrosion, and biofouling in brackish seawater: Comparison between six stainless steel grades, Bioelectrochemistry, 120, 27-42 (2018). Jia, R.; Yang, D.; Xu, D.; Gu, T.; Anaerobic Corrosion of 304 Stainless Steel Caused by the Pseudomonas aeruginosa Biofilm, Front Microbiol., 8, 2335 (2017). Sano, K.; Kanematsu, H.; Kogo, T.; Hirai, N.; Tanaka, T.; Corrosion and biofilm for a composite coated iron observed by FTIR-ATR and Raman spectroscopy, Transactions of the IMF, 94, 139-145 (2016).
Dwivedi, D.; Lepková, K.; Becker, T.; Carbon steel corrosion: a review of key surface properties and characterization methods, RSC Advances, 7, 4580-4610 (2017).
Xu, D.; Li, Y.; Gu, T.; Mechanistic modeling of biocorrosion caused by biofilms of sulfate reducing bacteria and acid producing bacteria, Bioelectrochemistry, 110, 52-58 (2016).
Kanematsu, H.; Barry, D. M. (Edi.) Corrosion Control and Surface Finishing: Environmentally Friendly Approaches. Tokyo: Springer Nature (2016)..
Melchers, R. E.; 9 - Modelling long term corrosion of steel infrastructure in natural marine environments, Liengen, T., Féron, D., Basséguy, R., Beech, I. B. (Edi.) Understanding Biocorrosion. 213-241, Oxford, Woodhead Publishing (2014).
Q3) Have the substrates been ground before applying the coatings?
Yes, the substrates were polished mechanically before applying the coating.
Q4) In page 7 (original) the authors wrote "Cu-coated coupons were covered with copper thermal sprayed coating but also showed some silver parts as well as red-brown-colored parts, which were probably the SS400 basal plate and iron rust" Where does this iron rust come from?
We thought this iron rust were made on the surface of the Cu-coated coupon. Unfortunately, Cu-coated coupon were very iron corrosive and it would suffer iron corrosion before the test. The reason is explained as below.
The surface of thermal spray coating is irregularly shaped and has gaps because the sprayed metals are aggregated and attached on the surface of a substrate. When moisture exists on the Cu-coated coupon, the coating element (Cu) and the substrate (Fe) are simultaneously immerged in a water solution. Then Fe become ionized state according to ionization tendency. Ionized irons move to the surface of Cu-coated coupon through the gaps of spray-coating, and combine to oxygen to iron oxides there (on the surface).
In addition, we added the following sentences after "Cu-coated coupons were covered ….. the SS400 basal plate and iron rust, respectively"
(Page 6, lines 6-13)
The iron rust would be made on the surface of the Cu-coated coupons. Because the Cu-coated coupons were very iron corrosive and it would suffer iron corrosion before the test. The surface of a thermal spray coated specimen is irregularly shaped and has gaps because the sprayed metals are aggregated and attached on the surface of a substrate. When moisture exists on the Cu-coated coupon, the coating element (Cu) and the substrate (Fe) are simultaneously immerged in a water solution. Then Fe become ionized state according to ionization tendency. Ionized irons move to the surface of Cu-coated coupon through the gaps of spray-coating, and combine to oxygen to iron oxides there.
Q5) Figure 9 is not correct. The figure depicts a continuous coating, but actually the coating, in both cases, as showed in figure 4, are full of pores. These pores allow the bacteria to move towards the substrate and, in the case of the cu coated, accelerating the corrosion process.
We re-drew Figure 9 to describe the irregularly coating shaped.
Q6) “It appeared that Cu-coated coupons were more aggressively corroded than SS400 coupons” was to be expected. The Cu-coated sample is not uniformly coated. In this case the water from either the external environment or bio-film penetrates between these pores and triggers a galvanic effect between the carbon steel and copper, leading to the deterioration of the substrate. Therefore, coatings applied in this way do more damage than good. since instead of protecting the substrate it accelerates the corrosion process.
With the Zn coated is the opposite.
We understood above comment as like we should explain what we expected Cu-coated and Zn-coated coupons. Therefore, we mentioned the reason why we performed the corrosion test of Cu and Zn thermal spray-coated SS400 in air and water environments, at the introduction section (page 2. lines 8-12).
Q7) Page 8, lines 1-2 (original): “To elucidate the relationship between corrosion progression and biofilm formation, sediments on the surfaces of outdoor-exposed coupons were identified as biofilms “. It is not clear. please clarify.
We deleted “To elucidate the relationship between corrosion progression and biofilm formation,”.
Q8) Page 10, lines 3-4 (original): Why “this result indicated that Zn-coated coupons had lower biofilm formation than SS400 and Cu-coated coupons”?
When a biofilm is stained using crystal violet, the color is changed to violet and the transparence is lower, i.e., chromaticity of a* is positive value, that of b* is negative value, and brightness (L*) is lower than 100. If a biofilm become thicker (mature), is be larger. Therefore, the amounts of biofilms can be quantified using the value of .
We thought above sentences helped readers in understanding the results. Therefore, we added them at page 9, lines 12-15, before “The value for Zn-coated coupons……”.
Q9) Page 11, lines 26--27 (original): “These findings implied that biofilms formed on the surface of all samples, however, MIC-related bacteria were rarely involved in the corrosion.” is not clear.
We rewrote this sentence as follow (page 11, lines 12-13):
These findings implied that MIC-related bacteria existed very low in the biofilms formed on these coupons and they were rarely involved in the corrosion.
Q10) Page 12, lines 2-3 (original): Why “the amount of biofilm on Zn-coated coupons was significantly lower than that on SS400 coupons”?
Page 17, lines 1-2 (original): Again, why? You must explain it. “Compared with the amount of biofilm formed on SS400, significantly more biofilm formed on Cu-coated coupons, while the amount of biofilm that formed on Zn-coated coupons was significantly lower (Figure 16).”
We added one explanation after the sentence ” This result indicated that Zn-coated coupons had lower biofilm formation than SS400 and Cu-coated coupons” (page 9, lines 18-26) as follows.
Kanematsu et al. has reported that Pseudomonas aeruginosa and Pseudoalteromonas carageenavara formed biofilms on the surface of SS400 much more than that of the other metal plated steels such as tin-, copper-, and zinc-plated ones, and iron (ion) could pull these bacteria better than other metals including tin, copper, and zinc [35]. In this study, iron oxide was detected on the surface of SS400 coupons and that of Cu-coated coupons after outdoor exposure test (Figure 5), on the other hand, zinc oxide was detected on the surface of Zn-coated coupons. Therefore, it was probably that the surface of SS400 coupons and that of Cu-coated coupons were rich of iron (ions) but that of Zn-coated coupons was poor of them, resulting in the difference of biofilm formation among these coupons.

Reviewer 2 Report
General comments:
It would be valuable to discuss practical aspects of the effects observed during experiments.
Section 4 "Conclusion" does not really concludes the main outcomes of the work. It looks more like a few general statements on results obtained. A more detailed description of the findings of the paper is needed.
Specific comments:
a) some of Figures must be improved, e.g. No 6, 9, 15 - some of denotations are very small and difficult to recognize, b) Tables 1-4: the [%] should be placed in the headings of the tables, c) Title of Table 5: -: not detected. (???) d) Figures 7 and 16: "P" parameter should be defined and discussed.
Author Response
Dear Reviewer,
Thank you very much for your kindly advices to improve the value of the article. We revised it according to your comments. We indicated the rewritten parts in red color.
In the original paper, we refer to SS400 carbon steel plate, zinc thermal spray-coated SS400 carbon steel plate, and copper thermal spray-coated SS400 carbon steel plate, as SS400, Zn-coating, and Cu-coating, respectively. However, we sometimes used “Cu-coated coupons” and “Zn-coated coupons” there. These mixing expressions were very confusing. Therefore, we conformed Cu-coating” and “Zn-coating” to “Cu-coated” and “Zn-coated” , respectively, including figures and tables.
We also transferred the following sentences to after “Cu-coated coupons was Sphingomonas (Table 3), while for Stramenopiles on Zn-coated coupons it was an unclassified new genus.” (page 18, lines 21-25) because we thought the position much better than before to understand our idea easily.
Additionally, we added corrosion story in a water environment about Zn-coated coupons and SS400 coupons at page 19, lines 9-15.
We hope it will become better and clearer scientific article than the original version.
<Questions and answers>
Q1) Section 4 "Conclusion" does not really concludes the main outcomes of the work. It looks more like a few general statements on results obtained. A more detailed description of the findings of the paper is needed.
We rewrote this section as below (page 20).
Conclusion
In this study, we investigated the effect of zinc thermal spray coated carbon steel (Zn-coated) and copper thermal spray coated carbon steel (Cu-coated) on iron corrosion in air or aquatic environmental condition. We also explored which factor(s) such as biofilm formation (MIC) and ionization tendency (electrochemical corrosion) worked dominantly in iron corrosion there. We expected Zn-coated and Cu-coated could inhibit iron corrosion due to sacrificial corrosion of zinc and contact killing of biofilm-forming bacteria, respectively. Both of air and water environments, Zn-coated inhibited iron corrosion but Cu-coated accelerated iron corrosion, which was negatively correlated to ionization tendency, i.e., Zn > Fe > Cu. The dominant factor of iron corrosion will be electrochemical corrosion in the air environment and MIC in the water environment, however, it is probably that biofilm formation plays an important role in iron corrosion. In the air environment, biofilms can store water (moisture) that makes galvanic cells to elute metallic ions according to ionization tendency. In the water environment, biofilm can accelerate iron corrosion caused by bacterial metabolites [67]. MIC-related bacteria have been found in specific environments such as oil tanks and water systems of nuclear power plants. In this study, MIC-related bacteria were barely detected in the biofilms formed on the surface of Zn-coated, Cu-coated, and carbon steel. This implies that biofilm formation is an essential factor for iron corrosion but MIC-related bacteria are not always necessary for it. Additionally, iron ions can predominantly attract bacteria more than zinc ions and copper ions, therefore, inhibiting elution of iron (ions) will be an effective approach to regulate biofilm formation as well as iron corrosion.
Q2) Some of Figures must be improved, e.g. No 6, 9, 15 - some of denotations are very small and difficult to recognize.
We changed the font sizes of axis titles of Figure 6 and 15. Additionally, we arranged Figure 9. We hope the new ones are easily recognized.
Q3) Tables 1-4: the [%] should be placed in the headings of the tables.
About Table 1, we explained meaning of the numbers in brackets in the title legend. About Table 2-4, [%] was placed in the headings.
Q4) Title of Table 5: -: not detected. (???)
This explanation was inappropriate. We replaced “-“ to “0” in the table, and deleted the sentence from the title.
Q5) Figures 7 and 16: "P" parameter should be defined and discussed.
We added the following sentences in these figure legends.
Each column shows the mean of modulus of (n = 5). Zn and Cu mean Zn-coated coupons and Cu-coated coupons, respectively. Error bars show standard deviation. Student’s t-test was performed between SS400 coupons and Zn-coated coupons or SS400 coupons and Cu-coated coupons. P means the p-value.

Reviewer 3 Report
In my opinion, the paper requires major revision. In order to improve this work Authors should take into account the following comments:
Page 4-5: the content of chapter 2.7 is identical to the one in (Antibiotics 2018, 7, 91), as it is experimental section, Authors may just refer to previous paper.
Page 7, Line 4-6: “while Cu-coated coupons were covered with copper thermal sprayed coating but also showed some silver parts … which were probably the SS400 basal plate.” It is well known that to protect steel against corrosion, a cathode (copper) or anode (zinc) coating can be used. However, in the first case, discontinuities (e.g. the pores mentioned by the authors on Page 7, line 18-20) cannot be tolerated and disqualifies such coating. Authors should explain why they chose such material.
Page 9, Line 12: the sentence “CH-NH-CH” is inconsistent with Figure 6. It should be corrected.
Page 10, Line 3-4: why “Zn-coated coupons had lower biofilm formation than SS400 and Cu-coated coupons” in the same corrosive environment? An explanation of the observed phenomenon should be given.
Page 14, Line 13-14: “Moreover, the amount of corrosive products on specimens immersed in the LBR was more than that for corrosive products of specimens exposed to the air.” How was the amount of corrosive products determined? Authors should provide specific values.
Figure 7 and 16: Description of the P parameter should be added in figure captions.
Page 20, Line 2: Zn-coating is a sacrificial coating, and after corrosion test is almost entirely covered by zinc oxide (it is shown in figures 3-5,10-12) thus, the sentence “Zn-coating was barely corroded at all” is incorrect and should be revised.
Page 20, Line 2-4: it should be expected that for the system porous copper coating/steel substrate the corrosion degree would be the highest (galvanic corrosion). However, in that system the steel substrate corrodes, not the copper coating, thus the sentence “corrosion tendency of Cu-coating, Zn-coating and SS400 (Cu-coating > SS400 > Zn-coating) was negatively correlated with the ionization tendencies of zinc, iron and copper (zinc > iron > copper)“ is incorrect and should be revised.
Page 20, Line 7: the sentence “electronic corrosion” is unclear and should be revised.
Page 20, Line 8-9: the sentence “while MIC was the dominant process in the water exposure tests” is inconsistent with the sentence “this result indicated that MIC-related bacteria were not involved in biofilm formation or corrosion of the coupons in the LBR environment” on page 19. It should be corrected.
Author Response
Dear Reviewer,
We appreciate your kindly comments for improving our manuscript. We revised it according to your comments as follows.
In the original paper, we refer to SS400 carbon steel plate, zinc thermal spray-coated SS400 carbon steel plate, and copper thermal spray-coated SS400 carbon steel plate, as SS400, Zn-coating, and Cu-coating, respectively. However, we sometimes used “Cu-coated coupons” and “Zn-coated coupons” there. These mixing expressions were very confusing. Therefore, we conformed Cu-coating” and “Zn-coating” to “Cu-coated” and “Zn-coated” , respectively, including figures and tables.
We also transferred the following sentences to after “Cu-coated coupons was Sphingomonas (Table 3), while for Stramenopiles on Zn-coated coupons it was an unclassified new genus.” (page 18, lines 21-25) because we thought the position much better than before to understand our idea easily.
Additionally, we added corrosion story in a water environment about Zn-coated coupons and SS400 coupons at page 19, lines 9-15.
We hope it will become better and clearer scientific article than the original version.
<Questions and answers>
Q1) Page 7, Line 4-6 (original): “while Cu-coated coupons were covered with copper thermal sprayed coating but also showed some silver parts … which were probably the SS400 basal plate.” It is well known that to protect steel against corrosion, a cathode (copper) or anode (zinc) coating can be used. However, in the first case, discontinuities (e.g. the pores mentioned by the authors on Page 7, line 18-20 (original) cannot be tolerated and disqualifies such coating. Authors should explain why they chose such material.
We explained the reason at introduction section (page 2. lines 8-12) as below.
The reason why we tested copper thermal spray coating was that we expected copper would postpone and(or) regulate MIC. Some researchers have reported that copper can inhibit microbial activities of some bacteria [13-15]. Additionally, we found that copper down-regulated biofilm formation of marine living bacteria [16].
Q2) Page 4-5 (original): The content of chapter 2.7 is identical to the one in (Antibiotics 2018, 7, 91), as experimental section. Authors may just refer to previous paper.
We changed chapter 2.7 as below, and added the reference. Additionally, deleted related references of original version.
<Revised version>
2.7. 16S rRNA Gene-Based Bacterial Community Analysis
The experimental procedure performed according to our previous study in Antibiotics [20].
<Added reference>
Ogawa, A.; Takakura, K.; Sano, K.; Kanematsu, H.; Yamano, T.; Saishin, T.; Terada, S.; Microbiome analysis of biofilms of silver nanoparticle-dispersed silane-based coated carbon steel using a next-generation sequencing technique. Antibiotics, 7, 91-100 (2018).
<Deleted references>
Edgar, R.C. Search and clustering orders of magnitude faster than BLAST. Bioinformatics, 26, 2460–2461 (2010).
Edgar, R.C. Drive5. Available online: http://www.drive5.com/ (accessed on 12 August 2018).
Caporaso, J.G.; Kuczynski, J.; Stombaugh, J.; Bittinger, K.; Bushman, F.D.; Costello, E.K.; Fierer, N.; Peña, A.G.; Goodrich, J.K.; Gordon, J.I.; et al. QIIME allows analysis of high-throughput community sequencing data. Nat. Methods 2010, 7, 335–336.
FASTX-Toolkit, FASTQ/A Short-Reads Pre-Processing Tools. Available online: http://hannonlab.cshl.edu/fastx_toolkit/ (accessed on 29 December 2019).
Joshi, N.A.; Fass, J.N. Sickle: A Sliding-Window, Adaptive, Quality-Based Trimming Tool for FastQ Files (Version 1.33) [Software]. 2011. Available online: https://github.com/najoshi/sickle (accessed on 29 December 2019).
Magoc, T.; Salzberg, S.L. FLASH: Fast length adjustment of short reads to improve genome assemblies. Bioinformatics, 27, 2957–2963 (2011).
Available online: FLASh Fast Length Ajustment of Short reads, https://ccb.jhu.edu/software/FLASH/ (accessed on 29 December 2019).
The Greengenes Database. Available online: http://greengenes.secondgenome.com (accessed on 29 December 2019).
Q3) The sentence “CH-NH-CH” is inconsistent with Figure 6. It should be corrected.
We revised the sentence as below (page 9, lines 4-5).
CH-NH-CH stretching vibration peaks associated with nucleic acids (2313-2500 cm-1 in SS400 coupon, 2290-2498 cm-1 in Cu-coated coupon, 2292–2500 cm-1 in Zn-coated coupon)
Q4) Page 10, Line 3-4 (original): why “Zn-coated coupons had lower biofilm formation than SS400 and Cu-coated coupons” in the same corrosive environment? An explanation of the observed phenomenon should be given.
We added one explanation after the sentence as follows (page 9, lines 18-26).
Kanematsu et al. has reported that Pseudomonas aeruginosa and Pseudoalteromonas carageenavara formed biofilms on the surface of SS400 much more than that of the other metal plated steels such as tin-, copper-, and zinc-plated ones, and iron (ion) could pull these bacteria better than other metals including tin, copper, and zinc [35]. In this study, iron oxide was detected on the surface of SS400 coupons and that of Cu-coated coupons after outdoor exposure test (Figure 5), on the other hand, zinc oxide was detected on the surface of Zn-coated coupons. Therefore, it was probably that the surface of SS400 coupons and that of Cu-coated coupons were rich of iron (ions) but that of Zn-coated coupons was poor of them, resulting in the difference of biofilm formation among these coupons.
Q5) Page 14, Line 13-14 (original): “Moreover, the amount of corrosive products on specimens immersed in the LBR was more than that for corrosive products of specimens exposed to the air”. How was the amount of corrosive products determined? Authors should provide specific values.
This sentence (Moreover, the amount of corrosive products on specimens immersed in the LBR was more than that for corrosive products of specimens exposed to the air.) was overstating, therefore, we deleted it and related sentences. Unfortunately, we do not have specific data about the amount of corrosive products. We measured the weight of each coupon before and after air exposure/LBR immersion tests to calculate corrosive products. But this method did not work because some parts of them were removed from the coupons by wind (in the air), water flow (in LBR).
<Deleted sentences>
Moreover, the amount of corrosive products on specimens immersed in the LBR was more than that for corrosive products of specimens exposed to the air. It is inferred that corrosion in an aquatic environment was promoted not only by electrochemical corrosion but also by microbially influenced corrosion (MIC).
Q6) Figure 7 and 16: Description of the P parameter should be added in figure captions.
We described the P parameter in the figure captions as follow.
Zn and Cu mean Zn-coated coupons and Cu-coated coupons, respectively. Error bars show standard deviation. Student’s t-test was performed between SS400 coupons and Zn-coated coupons or SS400 coupons and Cu-coated coupons. P means the p-value.
Q7) Page 20, Line 2 (original): Zn-coating is a sacrificial coating, and after corrosion test is almost entirely covered by zinc oxide (it is shown in figures 3-5,10-12) thus, the sentence “Zn-coating was barely corroded at all” is incorrect and should be revised.
We revised it to as below (page 19, lines 24-25).
while the Zn-coated coupons were almost entirely covered with zinc oxide, not iron oxide.
Q8) Page 20, Line 2-4 (original): it should be expected that for the system porous copper coating/steel substrate the corrosion degree would be the highest (galvanic corrosion). However, in that system the steel substrate corrodes, not the copper coating, thus the sentence “corrosion tendency of Cu-coating, Zn-coating and SS400 (Cu-coating > SS400 > Zn-coating) was negatively correlated with the ionization tendencies of zinc, iron and copper (zinc > iron > copper)“ is incorrect and should be revised.
Confusingly, Cu-coating and Zn-coating are name of coupons, not meaning of “coating”. We mentioned about it at section 2.1 (original). We changed the name of these coupons from Cu-coating” and “Zn-coating” to “Cu-coated coupons” and “Zn-coated coupons”, respectively, in revised version. Additionally, we rewrote the related parts as follows (page 19, lines).
This iron corrosion tendency of Cu-coated coupon, Zn-coated coupon and SS400 (Cu-coated coupon > SS400 coupon > Zn-coated coupon) was negatively correlated with the ionization tendencies of zinc, iron and copper (zinc > iron > copper).
Q9) Page 20, Line 7 (original): the sentence “electronic corrosion” is unclear and should be revised.
We revised it to electrochemical corrosion (page, lines 23-25).
Q10) Page 20, Line 8-9 (original): the sentence “while MIC was the dominant process in the water exposure tests” is inconsistent with the sentence “this result indicated that MIC-related bacteria were not involved in biofilm formation or corrosion of the coupons in the LBR environment” on page 19 (original). It should be corrected.
These sentences are corrected. Confusingly, MIC is different meaning from MIC-related bacteria. Generally, MIC is induced by MIC-related bacteria such as Pseudomonas, SRB such as Desulphovibrio, iron-oxidizing bacteria such as Acidthiobacillus(Thiobacillus), and Gallionella. However, in this study, MIC-related bacteria did not detect (list) in the biofilms formed on the surfaces of coupons.

Reviewer 4 Report
The manuscript “Biofilm formation plays a crucial rule in the initial step of carbon steel corrosion in air and water environments” presents the results of the evaluation of the effect of zinc coating on protecting carbon steel against biofilm formation in different environments. It is explained that zinc coating notably inhibited corrosion, biofilm formation in both tested environments. MIC-related bacteria are barely involved in the occurrence of corrosion.
The work is very well performed and reports interesting data related to an important issue, still some points, that need to be adjusted before publication, namely there are several missing details in M&M. long.
Introduction:
Page 2: “Several mechanisms for MIC have been proposed, as follows: Producing galvanic cells,…” – remove the uppercase letter from “Producing”;Material and Methods:
Brands and countries of manufacture companies are missing in for the acrylic plates and need to be included; Figure 2 could have more quality…; Please indicate the DNA final concentration for the amplification; Page 4: “2.7.16. S rRNA Gene-Based…” – 16S? Page 4: how were the primers designed? Reference or software links used? Page 4: for a higher sensitivity, the primers have approximately 15 bp. Why the chosen primers are so long? Please clarify; Page 5: “The 16S rRNA gene library…” – remove the italic form of 16S rRNA”; Page 5: indicate identify_chimeric_seqs.py script, Greengenes database, and pick_de_novo_otus.py script links; Page 5: “Raw data files have been deposited in NCBI Sequence Read Archive and are waiting the assignment of an accession number” – if it is already attributed, please add this information; Page 5: “Before and after exposure testing, the surface of each coupon was observed at 5-fold or 100-fold magnification using the attached microscope of a laser Raman spectroscope (NRS-3100, JASCO Co., Tokyo, Japan)” – full stop is missing; Page 5: “For before exposure testing,…” – this expression is not OK. “Before exposure testing,…” ? Page 5: “0.1% crystal violet solution for 30 min at 25 °C.” – 30 minutes? Normally is 5 minutes to avoid overstaining. Justification or reference for this change? Page 6: the CV protocol is altered. Wasn’t the biomass dissolved in acetic acid to measure the absorbance? Page 6: “Measured data were described using the L*a*b* color system: L represents lightness (calibration value was 100), a* represents the red/green coordinate (calibration value was zero), and b* represents the yellow/blue coordinate (calibration value was zero). If the color is violet, a* assumes a positive value and b* a negative value. We calculated the three-dimensional vector values, i.e., √(?∗)2 + (?∗)2 + (100 − ?∗)2 to infer the extent to which the sample formed a biofilm and indicating how a sample formed a biofilm.” – Reference is missing; Page 7: “exposure testing [Figure 5(a)]. Weld splashes of thermally sprayed materials, such as copper and zinc, were observed in Cu- and Zn- coated coupons before exposure testing [Figures 5 (b) and (c)]” - replace for “]” parenthesis; Page 9: Figure 8 – color 1, 5, 13, 17 and 23 are too similar. This reviewer advises to use other colors;Results/Discussion:
Page 19: please explore more the fact that “Methyloversatilis” is highly present in Zinc coated material (compare to possible existent past reports?); Page 19: “the following story was inferred: A small amount of copper” – replace the uppercase for lowercase letter in the sentence.Author Response
Dear Reviewer,
We appreciate your kindly comments about our current work. We revised the manuscript according to your comments. We indicated the rewritten parts in red color.
In the original paper, we refer to SS400 carbon steel plate, zinc thermal spray-coated SS400 carbon steel plate, and copper thermal spray-coated SS400 carbon steel plate, as SS400, Zn-coating, and Cu-coating, respectively. However, we sometimes used “Cu-coated coupons” and “Zn-coated coupons” there. These mixing expressions were very confusing. Therefore, we conformed Cu-coating” and “Zn-coating” to “Cu-coated” and “Zn-coated” , respectively, including figures and tables.
We also transferred the following sentences to after “Cu-coated coupons was Sphingomonas (Table 3), while for Stramenopiles on Zn-coated coupons it was an unclassified new genus.” (page 18, lines 21-25) because we thought the position much better than before to understand our idea easily.
Additionally, we added corrosion story in a water environment about Zn-coated coupons and SS400 coupons at page 19, lines 9-15.
We hope it will become better and clearer scientific article than the original version.
<Questions and answers>
Q1) Introduction, Page 2 (original) : “Several mechanisms for MIC have been proposed, as follows: Producing galvanic cells,…” – remove the uppercase letter from “Producing”;
We removed “Producing” from the sentence.
Q2) Material and Methods, Brands and countries of manufacture companies are missing in for the acrylic plates and need to be included;
We added the information.
Section 2.2 (page 2):
Each coupon was put on an acrylic plate (Sakai Netsu-Giken) with two corners held in place by stainless steel clamps (Sakai Netsu-Giken).
Section 2.3 (page 3):
SS400 and each coated coupon were cut into 2 cm × 1 cm rectangles using a shearing machine (Komatsu, Kanazawa, Japan). Non-coated sides were masked using silicone sealant (Cemedine Co. Ltd., Tokyo, Japan). We used an open laboratory biofilm reactor (LBR) [17] for aquatic biocorrosion accelerated testing (Figure 2). This open LBR was made by Sakai Netsu-Giken, and consisted of a cylindrical column, a water tank, an air fan and a pump.
Q3) Material and Methods, Figure 2 could have more quality…;
We added the photo image of the LBR and changed the legend.
Q4) Material and Methods, please indicate the DNA final concentration for the amplification;
We appreciate your kind comment. We would like to inform you that we arranged whole section 2.7 according to the comment of another reviewer. That is why this comment is not reflected the revised version.
The final DNA concentration of each sample was variable (< 0.01 ng/ml – 13.5 ng/ml) because each sample DNA concentration was different.
Q5) Material and Methods, page 4 (original): “2.7.16. S rRNA Gene-Based…” – 16S?
Yes, it is. We renamed the section title: 2.7. 16S rRNA Gene-Based Bacterial Community Analysis.
Q6) Material and Methods, page 4: how were the primers designed? Reference or software links used?
This next generation sequencing (NGS) was entrusted with Bioengineering lab (Kanagawa, Japan) and we cannot know about the information. However, we know about following information: the NGS was performed using Mi-Seq, illumina system (https://jp.illumina.com/systems/sequencing-platforms/miseq-fgx.html) and the microbiological taxonomy was analyzed according to QIIME program (http://qiime.org). Therefore, the primers were probably designed based on the platform of QIIME package.
Q7) Material and Methods, please clarify; Page 5 (original): “The 16S rRNA gene library…” – remove the italic form of 16S rRNA”.
Page 5: indicate identify_chimeric_seqs.py script, Greengenes database, and pick_de_novo_otus.py script links;
We appreciate your kind comments. We would like to inform you that we arranged whole section 2.7 according to the comment of another reviewer. That is why this comment is not reflected the revised version.
We list the links as below.
identify_chimeric_seqs.py script: http://qiime.org/scripts/
Greengenes database: http://greengenes.secondgenome.com/?prefix=downloads/greengenes_database/
pick_de_novo_otus.py: http://qiime.org/scripts/
Q8) Material and Methods, page 4 (original): for a higher sensitivity, the primers have approximately 15 bp. Why the chosen primers are so long?
These primers have barcode information that need for recognizing the target sample to analyze 16S RNA libraries.
Q9) Material and Methods, page 5 (original): “Raw data files have been deposited in NCBI Sequence Read Archive and are waiting the assignment of an accession number” – if it is already attributed, please add this information;
Unfortunately, it has not been yet.
Q10) Material and Methods, page 5 (original): “Before and after exposure testing, the surface of each coupon was observed at 5-fold or 100-fold magnification using the attached microscope of a laser Raman spectroscope (NRS-3100, JASCO Co., Tokyo, Japan)” – full stop is missing.
We added a full stop.
Q11) Material and Methods, page 5 (original): “For before exposure testing,…” – this expression is not OK. “Before exposure testing,…” ?
We changed these sentences as below (page 4, lines 31-34).
About non-exposure tested coupons, five points (in the vicinities of the center and the four corners) were randomly selected for each coupon. About exposure tested coupons, five spots where deposits were observed were selected for each coupon.
Q12) Material and Methods, page 5 (original): “0.1% crystal violet solution for 30 min at 25 °C.” – 30 minutes? Normally is 5 minutes to avoid overstaining. Justification or reference for this change?
Crystal violet staining performed 30 min. About biofilm staining, we have used this time and successfully detected the differences of the amount of biofilms. We added the reference of it.
<reference>
Ogawa, A.; Kiyohara, T.; Kobayashi, Y.; Sano, K.; Kanematsu, H.; Nickel, molybdenum, and tungsten nanoparticle-dispersed alkylalkoxysilane polymer for biomaterial coating: evaluation of effects on bacterial biofilm formation and biosafety. Biomedical Research and Clinical Practice, 2, 1-7 (2017)
Q13) Material and Methods, page 6 (original): the CV protocol is altered. Wasn’t the biomass dissolved in acetic acid to measure the absorbance?
No, it was not. The traditional CV protocol did not work for these thermal sprayed coating treated coupons because CV was remained on the roughly surface of them even if non-tested coupons. That is why we transferred the stained parts (biofilms) to the tape and measured using the colormeter.
Q14) Material and Methods, page 6 (original) : “Measured data were described using the L*a*b* color system: L represents lightness (calibration value was 100), a* represents the red/green coordinate (calibration value was zero), and b* represents the yellow/blue coordinate (calibration value was zero). If the color is violet, a* assumes a positive value and b* a negative value. We calculated the three-dimensional vector values, i.e., √(?∗)2 + (?∗)2 + (100 − ?∗)2 to infer the extent to which the sample formed a biofilm and indicating how a sample formed a biofilm.” – Reference is missing;
We added the reference:
Ogawa, A.; Kiyohara, T.; Kobayashi, Y.; Sano, K.; Kanematsu, H.; Nickel, molybdenum, and tungsten nanoparticle-dispersed alkylalkoxysilane polymer for biomaterial coating: evaluation of effects on bacterial biofilm formation and biosafety. Biomedical Research and Clinical Practice, 2, 1-7 (2017)
Q15) Results/Discussion, page 7 (original): “exposure testing [Figure 5(a)]. Weld splashes of thermally sprayed materials, such as copper and zinc, were observed in Cu- and Zn- coated coupons before exposure testing [Figures 5 (b) and (c)]” - replace for “]” parenthesis;
We replaced “[“ and “]” to “(“ and “)”, respectively.
Q16) Results/Discussion, page 9 (original): Figure 8 – color 1, 5, 13, 17 and 23 are too similar. This reviewer advises to use other colors;
We changed these colors. We hope the new colors are distinguishable.
Q17) Results/Discussion, page 19: please explore more the fact that “Methyloversatilis” is highly present in Zinc coated material (compare to possible existent past reports?);
We added more information as follow (underline, page 18, lines 27-29):
Methyloversatilis comprises three species which can utilize single carbon compounds, such as methanol and methylamine, as a sole source of energy [60–62].
Unfortunately, we could not find any past reports discussed about the occupancy of Methyloversatilis in environmental samples. The information about Methyloversatilis is just about bacterial characterization in the following three references.
Kalyuzhnaya, M. G., De Marco, P., Bowerman, S., Pacheco, C. C., Lara, J. C., Lidstrom, M. E., Chistoserdova, L.: Methyloversatilis universalis gen. nov., sp. nov., a novel taxon within the Betaproteobacteria represented by three methylotrophic isolates. Int J Syst Evol Microbiol, 56, 2517–2522 (2006)
Doronina, N. V., Kaparullina, E. N., Trotsenko, Y. A.: Methyloversatilis thermotolerans sp. nov., a novel thermotolerant facultative methylotroph isolated from a hot spring. Int J Syst Evol Microbiol, 64, 158–164 (2014)
Smalley, N. E., Taipale, S., De Marco, P., Doronina, N. V., Kyrpides, N., Shapiro, N., Woyke, T., Kalyuzhnaya, M. G.: Functional and genomic diversity of methylotrophic Rhodocyclaceae: description of Methyloversatilis discipulorum sp. nov. Int J Syst Evol Microbiol, 65, 2227–2233 (2015)
Q18) Results/Discussion, page 19 (original): “the following story was inferred: A small amount of copper” – replace the uppercase for lowercase letter in the sentence.
We changed it.

Round 2
Reviewer 1 Report
All questions were addressed satisfactorily. The manuscript can be published as it is.
Although I still have some doubts about figure 9. The figure does not describe a porous coating well enough.
Author Response
Dear Reviewer,
Thank you for your valuable comments and your precious time. Sorry that our explanation could not persuade you enough. We took your comments into our consideration and added some sentences for our explanation about Figure 9 (step 2) (blue colored sentences, at page 12, line 1-4).
Once again, thank you!
Reviewer 3 Report
In my opinion, the paper requires minor revision i.e. the point of question Q3 was that in the text is CH-NH-CH while in figure 6 is OH-NH-CH. Please check this.
Author Response
Dear Reviewer,
Thank you for your detailed check. We appreciate. We modified the text to OH-NH-CH (the changed letter shown in blue, page 9, line 4).